# Comparative Analysis of Hindgut Microbiota Variation in *Protaetia brevitarsis* Larvae across Diverse Farms

**DOI:** 10.3390/microorganisms12030496

**Published:** 2024-02-29

**Authors:** Namkyong Min, Jean Geung Min, Paula Leona T. Cammayo-Fletcher, Binh T. Nguyen, Dongjean Yim

**Affiliations:** 1Department of Applied Biology, Kyungpook National University, Daegu 41566, Republic of Korea; minnamkyong1@gmail.com (N.M.); jeanmin1@knu.ac.kr (J.G.M.); 2Institute of Animal Medicine, College of Veterinary Medicine, Gyeongsang National University, Jinju 52828, Republic of Korea; cammayopaula@gmail.com (P.L.T.C.-F.); thanhbinhcnty@gmail.com (B.T.N.); 3Hoxbio, Business Center, Gyeongsang National University, Jinju 52828, Republic of Korea

**Keywords:** *Protaetia brevitarsis*, third-instar larvae, hindgut microbiota, microbial profiling, microbiota variation

## Abstract

*Protaetia brevitarsis* larvae are farm-raised for food, are used in traditional East Asian medicine, and convert organic waste into biofertilizers. Here, the comparative analysis of the gut microbiota of third-instar larvae obtained from five different farms was investigated using 16S rRNA microbial profiling. Species richness, evenness, and diversity results using α-diversity analysis (observed species, Chao1, Shannon, Simpson) were similar between farms, except for those between the TO and KO farms. β-diversity was significantly different in distribution and relative abundance between farms (PERMANOVA, pseudo-F = 13.20, *p* = 0.001). At the phylum level, *Bacillota*, *Bacteroidota*, *Actinomycetota*, and *Pseudomonadota* were the most dominant, accounting for 73–88% of the hindgut microbial community. At the genus level, *Tuberibacillus, Proteiniphilum, Desulfovibrio, Luoshenia*, and *Thermoactinomyces* were the most abundant. Although oak sawdust was the main feed component, there were large variations in distribution and relative abundance across farms at the phylum and genus levels. Venn diagram and linear discriminant analysis effect size analyses revealed large variations in the hindgut microbial communities of *P. brevitarsis* larvae between farms. These results suggest environmental factors were more important than feed ingredients or genetic predisposition for the establishment of the intestinal microbiota of *P. brevitarsis* larvae. These findings serve as reference data to understand the intestinal microbiota of *P. brevitarsis* larvae.

## 1. Introduction

Insects and wild animals are often the main protein source for people who inhabit forest areas and tropical countries [1]. Insects grow and reproduce easily and have high feed conversion rates. They can be raised using feed sources, such as bio-waste and organic matter, which are then converted into high-value food and feed resources. Insect farming is one way to enhance food and feed security when insects are produced in a mass production system like livestock and developed into a palatable food [2]. About 2000 insect species are eaten in various parts of the world, mainly in tropical areas [1]. According to the European Food Safety Scientific Committee, in 2005, nine insect species were farmed for food and feed. *P. brevitarsis* (i.e., white-spotted flower chafer) is one edible insect species currently farmed for food production [3,4].

*P. brevitarsis*, a beetle species of the family Cetoniide of the order Coleoptera, is an important agricultural pest that is widely distributed in Europe and Northeastern Asia [5]. It is also used in China, Japan, and Korea as a traditional animal-based medicine. The drying yield of *P. brevitarsis* larvae significantly varies among commercial farms in Korea, ranging from 14.21% to 27.28% depending on differences in the period of fasting before harvest and/or the presence or absence of fasting with water immersion [6]. Dried *P. brevitarsis* larvae contain 46–54% protein and 13–26% fat content [7,8,9,10]. The high ratio of unsaturated fatty acids (80.54% of total fatty acids) to saturated fatty acids (19.46% of total fatty acids) is a distinctive feature of the fatty acid profile of *P. brevitarsis* larvae [8]. The most abundant fatty acid, oleic acid, accounts for more than 58% of the total fatty acids [9,11]. Although nutritional and functional chemical composition can be changed in response to feed materials and supplementary feeds [6,10,11], *P. brevitarsis* larvae also have a broad range of pharmacological activities, including anti-obesity and anti-diabetic effects [12,13], therapeutic effects on prostatic hyperplasia [14], the inhibition of osteoclastogenesis [15], neuroprotective effects against seizures and neurodegeneration [16], anti-bacterial and anti-inflammatory activities [17,18], and an anti-thrombotic function [19].

The gut microbiome is involved in intestinal homeostasis and host health by aiding intestinal absorption and fermentation, improving energy utilization, and combating various pathogenic microorganisms. Factors such as feed ingredients, supplementary feed, the farm environment, and genetics affect the intestinal microbial community composition [20,21,22,23]. In the case of insects, given the diversity in physiology, anatomy, morphology, and ecology, the gut microbial community is also likely to be highly diverse across species. Depending on the characteristics of the insect, some present specialized and dense intestinal microbial communities, while others present transient and relatively sparse communities acquired from the environment [20,23]. Even in mammals such as pigs, the fecal microbiota varies depending on growth stage and age [21,24]. However, compared with mammals, holometabolous metamorphosis is a phenomenon unique to insects and has major effects on the differences between insect larvae and adults [20,25].

We and others have hypothesized that feed ingredients can alter the nutritional composition of *P. brevitarsis* larvae [6,10,11]. Until now, the limited analyses of microorganisms have been conducted for the gut contents of *P. brevitarsis* larvae. Therefore, information needed for research, such as for comparisons between farms, is lacking [26,27]. Here, we analyzed the hindgut bacterial communities of *P. brevitarsis* larvae from different insect farms based on 16S rRNA gene sequence analysis. We also performed a comparative analysis of the gut microbiota present.

## 2. Materials and Methods

### 2.1. Insects and Sampling

Third-instar larvae of *P. brevitarsis* were obtained from five farms in Korea. Three farms (KB, BR, IS) fed commercial feed; two farms (TO, JH) fed homemade feed. Except for BR farm, which mainly used spent mushroom substrates, the feed for four of the farms included oak sawdust as the main feed ingredient. The mean larval body weight exceeded 2 g per larva (Table 1). To obtain hindgut samples, larvae were immersed in 75% alcohol and subjected to three washes with phosphate-buffered saline (PBS) [28]. The digestive tract of the five larvae was dissected in each procedure to extract gut contents, and about 500 mg of hindgut contents were collected into a 2 mL sterile plastic tube. The collected samples were stored at −80 °C until microbial DNA extraction and sequencing was performed.

### 2.2. DNA Preparation and Sequencing

Total metagenomic DNA was extracted from hindgut samples using a fecal DNA isolation kit (MO BIO, Carlsbad, CA, USA), following the manufacturer’s instructions. DNA concentrations were measured using a QuantiFluor dsDNA system (Promega, Madison, WI, USA) and a Victor Nivo Multimode Microplate Reader (PerkinElmer, Waltham, MA, USA). The hypervariable V3–V4 region of the microbial 16S rDNA gene was amplified using polymerase chain reaction (PCR) with Herculase II Fusion DNA polymerase and a Nextera XT DNA index V2 kit with an Illumina adapter containing a primer set (Bakt 341F/805R primers) based on 16S Metagenomic Sequencing Library Preparation (Illumina, San Diego, CA, USA). The PCR conditions were 3 min at 95 °C for heat denaturation and 25 cycles of 30 s at 95 °C, 30 s at 55 °C, and 30 s at 72 °C, followed by a final 5 min extension at 72 °C. Samples between 600 and 620 bp were chosen for sequencing. The prepared amplicons were sequenced using the Illumina MiSeq platform and a 301 bp paired-end format (Macrogen, Seoul, Republic of Korea).

### 2.3. Bioinformatics and Sequencing Data Processing

The 16S amplicon sequences were processed using a Divisive Amplicon Denoising Algorithm (DADA2). Diversity and taxonomic analyses were performed using amplicon sequence variant (ASV) sequences. Adaptor and primer trimmings on raw leads were performed using Cutadapt (https://cutadapt.readthedocs.io/en/v3.2/ accessed on 20 November 2023); DADA2 was used for qualify filtering, denoising, merging, and chimera removal [29]. Taxonomic assignments were determined using BLAST (version 2.9.0) against the NCBI 16S microbial database. Unless otherwise indicated, software default settings were used throughout the analysis [30].

The α-diversity of the groups, i.e., observed species and Chao1, Shannon, and Simpson index values, was evaluated. The β-diversity between samples was estimated using Bray–Curtis dissimilarities and visualized using classical-metric multidimensional scaling. Statistical significance between groups was assessed using a permutational analysis of variance (PERMANOVA) test (QIIME 2 bioinformatics platform). Venn diagram analysis among groups was performed on all genus levels and visualized using Python 3 (version 3.6.2).

Cladogram and linear discriminant analysis (LDA) coupled with an effect size (LEfSe) algorithm (http://huttenhower.sph.harvard.edu/ accessed on 4 January 2024) was performed to identify significant among-group microbial differences. Taxa with significant differential abundance were detected using nonparametric factorial Kruskal–Wallis rank sum tests. A significance value of <0.05 and an LDA effect size of >4 were used as thresholds for the LEfSe analysis.

### 2.4. Statistical Analysis

The data analysis was performed using the statistical software packages R (version 3.6.2) and Instat (GraphPad, La Jolla, CA, USA). Nonparametric statistical tests were chosen for all analyses. The Kruskal–Wallis H test was performed with a post hoc analysis using Dunnett’s multiple comparisons test. Results were presented as mean ± standard deviation values. Statistical significance was defined as *p* < 0.05.

## 3. Results

### 3.1. General Information and Sample Sequencing

Twenty-five hindgut samples of third-instar *P. brevitarsis* larvae collected from five different farms across Korea (Table 1) were sequenced. The detailed sequencing data for each sample are presented in Appendix A. The median read of the original total reads was 156,346 (range: 126,596 to 177,274) and that of the highly qualified reads used for the final analysis was 67,870 (range: 54,210 to 77,960). The median was 86.8% (range: 83.5% to 88.5%) (Appendix A). A total of 21,484 ASVs were obtained from 25 samples; the median number was 853 ASVs (range: 619 to 1207). The Good’s coverage index was >99%. The Good’s coverage, Shannon–Wiener, Chao1, and rarefaction curve values for the samples indicated that there was sufficient data sampling and adequate sequencing depth, and the database of 16S rRNA gene-based sequences covered almost all microbial communities (Figure 1 and Appendix A).

### 3.2. Diversity Analyses and Gut Microbial Community

Alpha diversity represents species richness, evenness, and diversity. Microbial diversity between the TO and KB groups showed significant variation based on the observed ASV (*p* < 0.05), Chao1 (*p* < 0.05), and Shannon (*p* < 0.01) index results. The TO and KB farms had the highest and the lowest values, respectively, compared with those of other farms (Figure 1A–C). However, Simpson index results indicated there were no significant between-group differences (Figure 1D). Principal coordinate analysis with taxonomy abundance of species was performed using the Bray-Curtis distance statistic (Figure 1E). Principal component 1 and principal component 2 explained 31.75% and 20.98%, respectively, of the variance of variables; the cumulative contribution rate was up to 52.73%. The resulting plot revealed that β-diversity between the five farms was significantly different (PERMANOVA, pseudo-F = 13.20, *p* = 0.001).

### 3.3. Analysis of Microbial Abundance and Composition

To characterize the microbial taxonomic distribution in *P. brevitarsis* larvae, we analyzed hindgut microbiota at the phylum and genus levels (Figure 2). At the phylum level, *Bacillota*, *Bacteroidota*, *Actinomycetota*, and *Pseudomonadota* were the most dominant phyla. The average abundances of *Bacillota, Bacteroidota, Actinomycetota*, and *Pseudomonadota* were 46.8% (range: 31.8% to 56.3%), 17.6% (range: 12.9% to 22.0%), 12.2% (range: 7.7% to 17.2%), and 7.5% (range: 4.0% to 11.0%), respectively (Figure 2A, Appendix A).

At the genus level, *Tuberibacillus*, *Proteiniphilum*, *Desulfovibrio*, *Luoshenia*, and *Thermoactinomyces* were the most abundant genera. The average abundances of *Tuberibacillus, Proteiniphilum, Desulfovibrio, Luoshenia,* and *Thermoactinomyces* were 6.4% (range: 0.2% to 20.7%), 6.2% (range: 4.8% to 9.0%), 4.0% (range: 2.6% to 8.7%), 3.4% (range: 1.3% to 4.9%), and 3.2% (range: 0% to 7.8%), respectively (Figure 2B, Appendix A).

*Streptomyces, Bacillus*, and *Pseudomonas*, which are considered agriculturally beneficial microorganisms [28], were monitored at the genus level. *Bacillus* was the most abundant (7.03%) at the KB farm. At the other four farms, *Bacillus* ranged from 0.44% to 1.67%. *Streptomyces* had the highest abundance (3.96%) at the IS farm; its abundance at the remaining four farms ranged from 1.28% to 2.42%. The relative among-farm abundance of *Bacillus* and *Streptomyces* were significantly different (Figure 3B,C). *Pseudomonas* was not detected or was found at very low levels (0.01% at the BR and IS farms) (Figure 3, Appendix A).

Differences between farms were analyzed across 10 phyla and 26 genera (Figure 4). Significant differences (Kruskal–Wallis H test, *p* < 0.05) were found for all phyla, except *Deferribacterota* (Figure 4A). Of the 26 genera analyzed, 22 (except *Proteiniphilum, Alistipes, Marasmitruncus*, and *Gehongia*) were found to be significant in terms of relative abundance, based on the Kruskal–Wallis H test result (*p* < 0.05) (Figure 4B).

### 3.4. Shared and Unique Microbial Populations

To compare the similarity of microbial composition among the five farms, Venn diagram analyses were performed at the genus level (Figure 5, Appendix A). Of the 812 genera, 185 were shared by all five farms. Overall, 48, 129, 51, 42, and 57 genera were unique to the KB, TO, BR, IS, and JH farms, respectively (Figure 5A). Among the 25 most abundant genera, the genus *Adhaeretor* was unique to the TO farm, *Dehalbactor* to the BR farm, and *Falsibacillus* to the JH farm; the KB and IS farms had no unique genera (Appendix A). When the KB and IS farms fed commercial feed containing oak sawdust as the main ingredient, 269 of 536 genera were shared; 159 and 108 genera were unique to the KB and IS farms, respectively (Figure 5B). As indicated in Figure 3C, when the TO and JH farms fed homemade feed containing oak sawdust as the main ingredient, 250 of 628 genera were shared; 254 genera were unique to the TO farm, and 124 genera were unique to the JH farm. Regardless of the feeding of commercial and homemade feeds containing oak sawdust as the main ingredient, these results indicated that there were between-farm differences in the hindgut microbiome composition in third-instar *P. brevitarsis* larvae.

### 3.5. Differential Bacterial Abundance

We performed LEfSe to generate a Cladogram and a Histogram of LDA scores to identify microbial taxa with significant differential abundance values across different farms (Figure 6, Figure 7 and Figure 8). A significance value of <0.05 and an LDA effect size of >4 were used as thresholds for the LEfSe analysis.

Overall, 76 differential microbial enrichments were detected between all five farms: 6 from BR, 18 from IS, 19 from JH, 9 from KB, and 24 from TO. These different abundances occurred mainly in the phyla *Bacillota*, *Actinomycetota*, and *Pseudomonadota* (Figure 6). In the case of the KB and IS farms, which fed commercial feed containing oak sawdust as the main ingredient, the differential enrichment of 24 microorganisms was detected in both groups (16 from IS, 7 from KB). The different abundances occurred in the phyla *Bacillota* and *Pseudomonadota* (Figure 7). When the KB and IS farms fed homemade feed containing oak sawdust as the main ingredient, the differential enrichment of 52 microorganisms was detected in both groups (25 from JH, 27 from TO). Although these different abundances occurred in eight phyla, the different abundances occurred mainly in three phyla, *Bacillota*, *Actinomycetota*, and *Pseudomonadota* (Figure 8).

## 4. Discussion

Agricultural wastes mainly originate from a variety of sources, including plant cultivation, livestock farming, and aquaculture. The proper use of the biomass associated with these wastes can reduce pollution and social concerns [31,32]. Edible insect *P. brevitarsis* larvae can act as important biomass decomposers, reducing waste by converting herbaceous and woody plant residues into humic acids and can thereby convert plant waste or other organic waste into proteins, fats, or other compounds that insects can use [31,32]. Biomass digested using *P. brevitarsis* larvae converts herbaceous (maize straw) and ligneous (sawdust) plant residues into organic fertilizers with a high humic acid content [33,34]. Spent mushroom substrate is a soil-like material left after mushroom cultivation. By fermenting this substrate with the frass of *P. brevitarsis* larvae or microbiota derived from animal manure (e.g., chicken manure), its usefulness for growing *P. brevitarsis* larvae can be increased [26]. Microorganisms in the frass of *P. brevitarsis* larvae have bio-stimulant activities, such as disease resistance and plant growth-promoting functions. The high-throughput microbial sequence analysis of samples related to *P. brevitarsis* larvae confirmed that the frass of *P. brevitarsis* larvae can affect the microbial composition of the surrounding environment [28]. Thus, frass can then be used to affect the microbial community of the surrounding environment [28]. Although a mechanistic study of lignocellulose degradation in the digestive tract of *P. brevitarsis* larvae was performed using the transcriptome analysis of midgut and hindgut tissues and the microbial community analysis of midgut and hindgut contents [27], the intensive analysis of the hindgut microbiota of *P. brevitarsis* larvae has been limited. Here, we used comparative analysis and investigated the hindgut microbial communities of *P. brevitarsis* larvae from five different farms using 16S rRNA microbial profiling.

In this study, α-diversity analysis results of 25 samples from five farms showed an average of 859 ± 133.7 ASVs, an average of 866 ± 136.9 for Chao1 values, and an average of 7.99 ± 0.4 for the Shannon and 0.99 ± 0.01 for the Gini–Simpson indices (Appendix A). Xuan et al. [28] found that the hindgut contents of *P. brevitarsis* larvae grown on spent mushroom substrates had an average of 2978 ± 195 operational taxonomic units (OTUs), an average of 3296 ± 158 for Chao1 value, an average of 8.93 ± 0.3 for Shannon index, and an average of 0.01 ± 0.004 for Simpson’s index. The hindgut samples of *P. brevitarsis* larvae had an average OTU value of 1621 ± 12 [35]. Taken together, these results indicated that species richness (or observed traits) can vary significantly between study results. Thus, growing conditions such as feed and environmental conditions can affect species richness.

Species richness, evenness, and diversity measured using α-diversity analysis were similar between farms, except for the differences between the TO and KO farms. However, based on β-diversity values, there were significant between-farm differences in distribution and relative abundance (PERMANOVA, pseudo-F = 13.20, *p* = 0.001). At the phylum level, the average abundance values for *Bacillota* (syn, *Firmicutes*), *Bacteroidota* (syn, *Bacteroidetes*), *Actinomycetota* (syn, *Actinobacteria*), and *Pseudomonadota* (*Proteobacteria*) were 46.8% (31.8% to 56.3%), 17.6% (12.9% to 22.0%), 12.2% (7.7% to 17.2%), and 7.5% (4.0% to 11.0%), respectively. This result indicated that the most dominant phyla varied significantly between farms. These four most dominant phyla accounted for 73% to 88% of the hindgut microbial communities. Similarly, one study found that the phyla *Firmicutes*, *Bacteroidetes*, *Proteobacteria*, and *Actinobacteria* are the predominant bacteria in the hindgut, accounting for >80% of the hindgut microbial communities of *P. brevitarsis* larvae [27]. In another study, *Proteobacteria, Bacteroidetes, Actinobacteria*, and *Firmicutes* were the most dominant phyla; they accounted for approximately 80 to 90% of the hindgut microbial community of *P. brevitarsis* larvae [28]. Interestingly, *Firmicutes*, known to be one of the key taxa associated with the reduction in antibiotic resistance genes [36], was shown to have differences in relative abundance. *Firmicutes* was the most abundant phylum in our study and the report [27] but was found to be the fourth most abundant phylum in another study [28]. Compared with previous studies [28], the most abundant bacterial communities at the genus level also differed in our study. These results indicated that microbial composition can vary greatly between the hindgut samples of *P. brevitarsis* larvae.

Venn diagram analyses performed at the genus level found that 185 of the 812 genera were shared by all five farms; 48, 129, 51, 42, and 57 genera were unique to the KB, TO, BR, IS, and JH farms, respectively. The KB and IS farms fed commercial feed containing oak sawdust as the main ingredient, and out of 536 genera, 159 were unique to the KB farm and 108 were unique to the IS farm. At the TO and JH farms that fed homemade feed containing oak sawdust as the main ingredient, of 628 genera, 254 were unique to the TO farm and 124 were unique to the JH farm. Similar results were obtained using the LEfSe analysis. Although oak sawdust and spent mushroom substrate are the main feed ingredients used on farms, the usefulness of these ingredients can be further increased by fermenting them based on the farm’s know-how including special additives (e.g., rice straw, rice bran, microbial agent), temperature, humidity, and mixing ratio. In addition, the rearing environment conditions due to differences in temperature, humidity, ventilation, and the particle size of soil are believed to affect the composition of the intestinal microorganisms of *P. brevitarsis* larvae. Considered collectively, these results suggested that the microbial community composition in the hindgut of third-instar larvae of *P. brevitarsis* varied depending on farm conditions and might affect the microbiota of the frass from these larvae. *P. brevitarsis* larvae-related samples that include the frass contain the agriculturally beneficial genera *Streptomyces*, *Bacillus*, and *Pseudomonas* [28]. In this study, we found differences between *Streptomyces* and *Bacillus* (but not *Pseudomonas*) and significant between-farm differences.

## 5. Conclusions

The reuse of agricultural wastes, such as spent mushroom substrate and chicken manure, is an important social issue. *P. brevitarsis* larvae, one of the edible insects, can be used to help reuse agricultural waste through methods such as increasing palatability and digestibility or the reduction in antibiotic-resistant bacteria present in chicken manure [26,37]. This study found significant differences between farms in the composition of the gut microbiota of *P. brevitarsis* larvae by comparative analysis. Therefore, it is important to first understand the composition of intestinal microorganisms of *P. brevitarsis* larvae for various studies such as agricultural waste recycling, larval farming, traditional medicine, or biofertilizer production using the gut microorganisms of *P. brevitarsis* larvae. Additional clarification and investigation are needed to determine whether different microbial compositions affect agricultural waste recycling results.

## Figures and Tables

**Figure 1 microorganisms-12-00496-f001:**
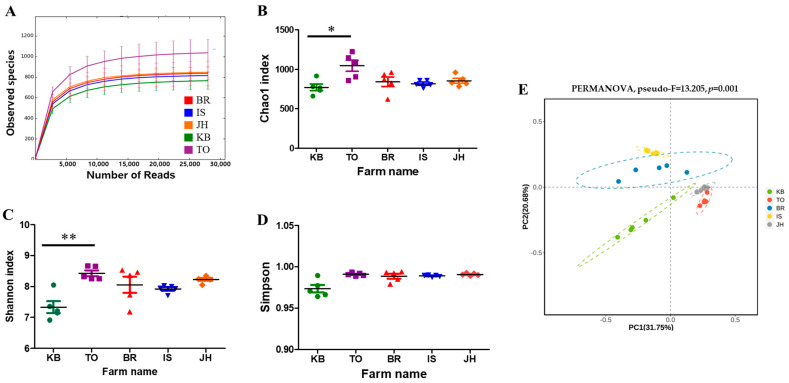
Diversity analysis of hindgut microbiota of third-instar larvae of *Protaetia brevitarsis.* (**A**) Rarefaction curves for observed species. (**B**) Chao1 index. (**C**) Shannon index. (**D**) Gini-Simpson index. The Kruskal-Wallis H test was used to assess whether there was a significant difference between farms. * *p <* 0.05 and ** *p <* 0.01 indicate statistically significant differences. (**E**) β-diversity, visualized using multidimensional scaling on Bray-Curtis dissimilarity. *p*-value derived from the PERMANOVA test with 999 permutations. KB, Kyungpook farm; TO, Tohamsan Gumbengi farm; BR, Gumbengi brothers; IS, Secomnalagum Gumbengi; JH, JHbio.

**Figure 2 microorganisms-12-00496-f002:**
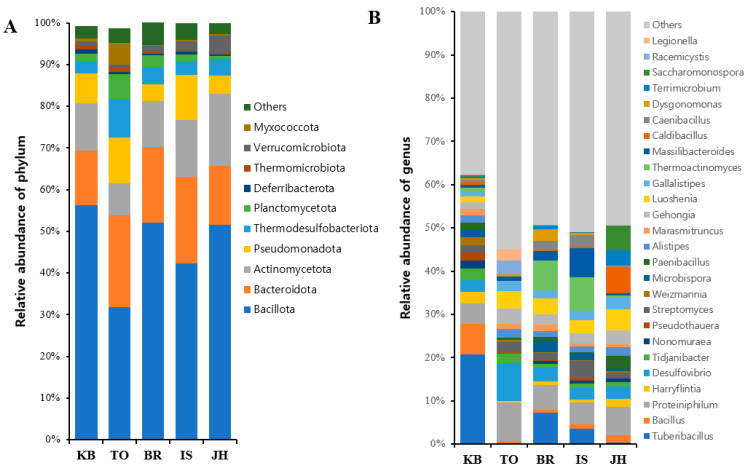
Gut microbial composition at phylum and genus levels. (**A**) Each bar represents the average relative abundance of each phylum of microbial community. (**B**) Each bar represents the average relative abundance of each genus. The community bar plots indicate percentage abundance at different taxonomic levels.

**Figure 3 microorganisms-12-00496-f003:**
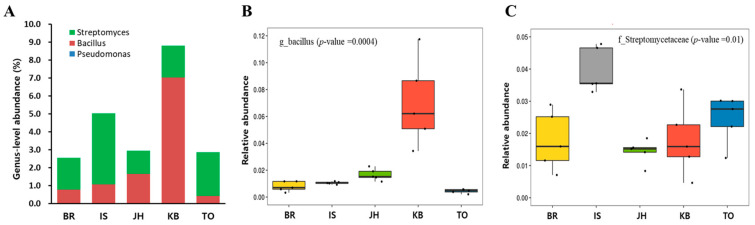
Abundance of *Streptomyces*, *Bacillus*, and *Pseudomonas* at the genus level. (**A**) Community bar plots indicate percentage abundance at different taxonomic levels. (**B**,**C**) Each bar represents the average relative abundance of *Bacillus* and *Streptomyces*. Dots indicate samples.

**Figure 4 microorganisms-12-00496-f004:**
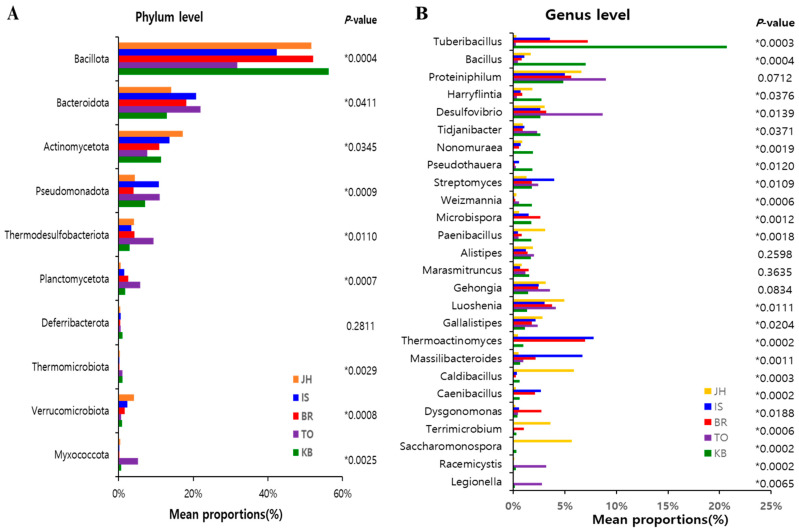
Distribution and relative abundance of 10 microbial phyla (**A**) and top 26 genera (**B**) assignable to ASV in different farm samples. The Kruskal-Wallis H test was used to assess significant differences (* *p <* 0.05). ASV, amplicon sequence variants.

**Figure 5 microorganisms-12-00496-f005:**
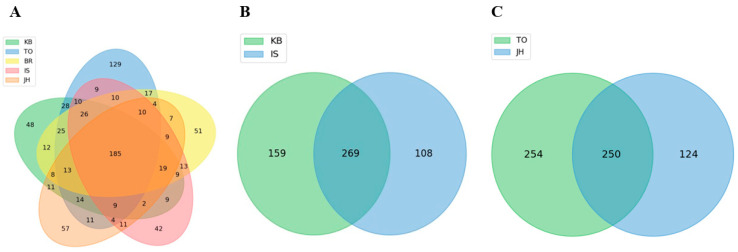
Venn diagram plots showing the ASV numbers of unique and shared genera. (**A**) Plot of the five farms. (**B**) Plot of the KB and IS farms. (**C**) Plot of the TO and JH farms. ASV, amplicon sequence variants.

**Figure 6 microorganisms-12-00496-f006:**
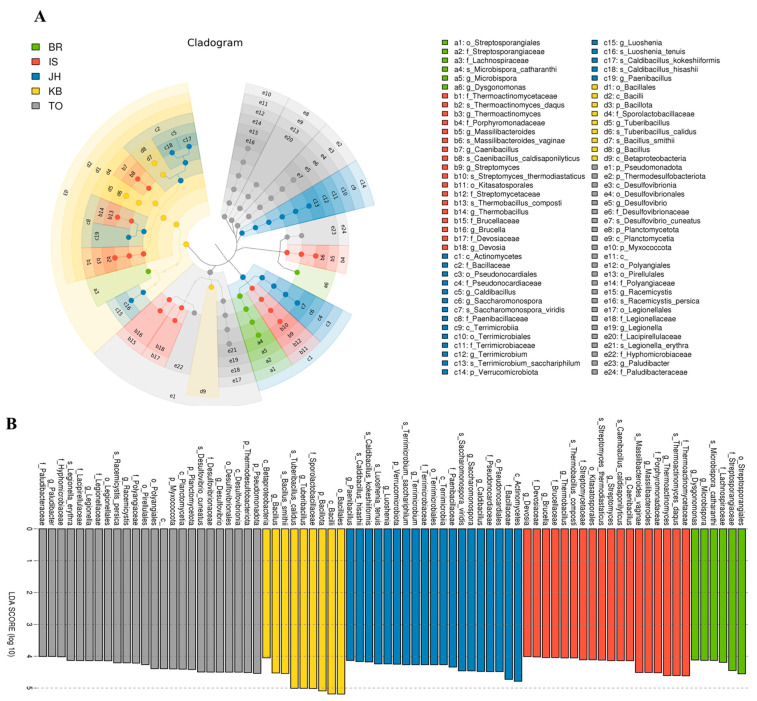
Different abundances of bacterial communities by Cladogram (**A**) and LDA (**B**) of LEfSe analysis. Histograms of the LDA scores computed for differentially abundant bacterial taxa among the BR, IS, JH, KB, and TO farms. A significance value of less than 0.05 and an LDA effect size of greater than 4 were used as thresholds for the LEfSe analysis. LEfSe, linear discriminant analysis effect size; LDA, linear discriminant analysis.

**Figure 7 microorganisms-12-00496-f007:**
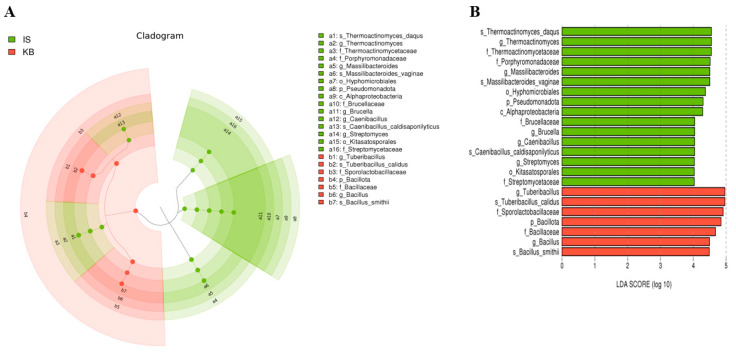
Different abundances of bacterial communities by Cladogram (**A**) and LDA (**B**) of LEfSe analysis. Histograms of the LDA scores computed for differentially abundant bacterial taxa between the IS and KB farms. A significance value of less than 0.05 and an LDA effect size of greater than 4 were used as thresholds for the LEfSe analysis. LEfSe, linear discriminant analysis effect size; LDA, linear discriminant analysis.

**Figure 8 microorganisms-12-00496-f008:**
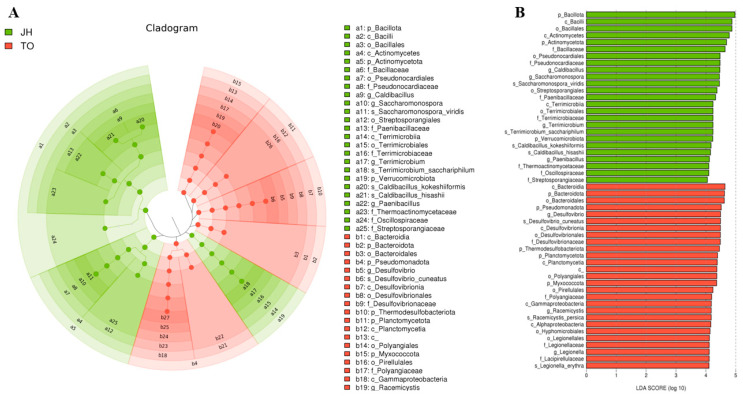
Different abundances of bacterial communities by Cladogram (**A**) and LDA (**B**) of LEfSe analysis. Histograms of the LDA scores computed for differentially abundant bacterial taxa between the JH and TO farms. A significance value of less than 0.05 and an LDA effect size of greater than 4 were used as thresholds for the LEfSe analysis. LEfSe, linear discriminant analysis effect size; LDA, linear discriminant analysis.

**Table 1 microorganisms-12-00496-t001:** General characteristics of white-spotted flower (*Protaetia brevitarsis*) larvae produced at commercial insect farms.

Farm Name	Farm Location	Feed	Main Ingredients	* Larval Weight (g)
KB	Gyeongsangbuk-do	commercial	oak sawdust	2.71 ± 0.31
TO	Gyeongsangbuk-do	homemade	oak sawdust	2.29 ± 0.15
BR	Incheon	commercial	spent mushroom substrate	2.66 ± 0.36
IS	Incheon	commercial	oak sawdust	2.48 ± 0.21
JH	Chungcheongnam-do	homemade	oak sawdust	2.03 ± 0.16

* Mean ± standard deviation (SD). KB, Kyungpook farm; TO, Tohamsan Gumbengi farm; BR, Gumbengi brothers; IS, Secomnalagum Gumbengi; and JH, JHbio.

## Data Availability

The data presented in this study are available on NCBI Sequence Read Archive, SAMN39848665-SAMN39848689.

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
