# Peer review of "Comparative Analysis of Hindgut Microbiota Variation in Protaetia brevitarsis Larvae across Diverse Farms"

_microorganisms, 2024, doi:10.3390/microorganisms12030496_

Round 1
Reviewer 1 Report
Comments and Suggestions for Authors
The manuscript “Comparative Analysis of the Hindgut Microbiota of Protaetia brevitarsis Larvae Grown at Different Farms” deals with the comparative analysis of the gut microbiota of third-instar larvae obtained from five different farms was investigated using 16S rRNA microbial profiling. The topic discussed is interesting and important as far as the Protaetia brevitarsis larvae are farm-raised for food and traditional East Asian medicine. I would like to make a few comments:
1) lines 22-24: “These results suggest environmental factors were more important than feed ingredients or genetic predisposition for establishment of the intestinal microbiota of P. brevitarsis larvae.”
Comment: Indicate and name please what environmental factors are implied.
2) lines 22-24: “Dried P. brevitarsis larvae contain 54-46% protein and 13-26% fat content [7-10]”.
Do microorganisms in the hindgut of larvae retain their viability? Can these organisms, when used in larval food, influence changes in human microflora?
3) When "homemade feed" is mentioned, what additional ingredients besides oak sawdust might be included? The discussion indicates bird droppings as a possible ingredient, in which case the question arises whether antibiotics were used in raising the poultry, or whether the droppings contained antibiotics that could affect the research results.
4) line 307-309: “These results suggested that the microbial community composition in the hindgut of third-instar larvae of P. brevitarsis varied depending on farm conditions and might affect the microbial flora of the frass from these larvae.”
Comment: Indicate and name please what farm conditions are implied.
5) line 245: one dot should be removed.
6) Larval microorganisms may contain antimicrobial peptides (bacteriocins), which can affect members of human microbiological communities. It is necessary to mention this probability in the discussion and quote sone of the latest publications, for example:
Simons, A.; Alhanout, K.; Duval, R.E. Bacteriocins, Antimicrobial Peptides from Bacterial Origin: Overview of Their Biology and Their Impact against Multidrug-Resistant Bacteria. Microorganisms 2020, 8, 639. https://doi.org/10.3390/microorganisms8050639
Guryanova, S.V. Immunomodulation, Bioavailability and Safety of Bacteriocins. Life 2023, 13, 1521. https://doi.org/10.3390/life13071521
7) In the discussion or conclusion, it is necessary to indicate what effect the microorganisms found in the larvae can have on the human body.
Author Response
We very much appreciate the constructive comments and suggestions of the reviewers, and are grateful, for the thorough examination of our data and recommendations for improvement. Below are our point-for-point responses. We hope these will meet with your approval.
Reviewer #1: Comments and Suggestions
The manuscript “Comparative Analysis of the Hindgut Microbiota of Protaetia brevitarsis Larvae Grown at Different Farms” deals with the comparative analysis of the gut microbiota of third-instar larvae obtained from five different farms was investigated using 16S rRNA microbial profiling. The topic discussed is interesting and important as far as the Protaetia brevitarsis larvae are farm-raised for food and traditional East Asian medicine. I would like to make a few comments:
1) lines 22-24: “These results suggest environmental factors were more important than feed ingredients or genetic predisposition for establishment of the intestinal microbiota of P. brevitarsis larvae.”
Comment: Indicate and name please what environmental factors are implied.
Response: The following sentence “Although oak sawdust and spent mushroom substrate are the main feed ingredients used on farms, the usefulness of these ingredients can be further increased by fermenting them based on the farm's know-how including special additives (e.g., rice straw, rice bran, microbial agent), temperature, humidity and mixing ratio. In addition, the rearing environment conditions due to differences in temperature, humidity, ventilation and particle size of soil are believed to affect the composition of the intestinal microorganisms of P. brevitarsis larvae” was added in line 313-319.
2) lines 22-24: “Dried P. brevitarsis larvae contain 54-46% protein and 13-26% fat content [7-10]”.
Do microorganisms in the hindgut of larvae retain their viability? Can these organisms, when used in larval food, influence changes in human microflora?
Response: Microorganisms in the hindgut of larvae have viability. Larvae is commonly taken in a dry form, a powered form, small pills, and/or a juicy form. It is believed that most or all microorganism will die during this process. Additionally, it is believed that it will have little or no effect on the human intestinal microflora due to destruction by stomach acid during the eating process.
3) When "homemade feed" is mentioned, what additional ingredients besides oak sawdust might be included? The discussion indicates bird droppings as a possible ingredient, in which case the question arises whether antibiotics were used in raising the poultry, or whether the droppings contained antibiotics that could affect the research results.
Response: At homemade feed, the usefulness of these ingredients can be further increased by fermenting them based on the farm's know-how including special additives (e.g., rice straw, rice bran, microbial agent), temperature, humidity and mixing ratio. The related sentence was added in line 315.
Generally, diluted frass of P. brevitarsis larvae has been used as microbial agent for re-fermentation. To find out other microbial agents, chicken manure was used, but it was not as effective as frass of P. brevitarsis. In addition, because chicken manure contains antibiotics, some research is focused on reducing antibiotic-resistant bacteria using P. brevitarsis larvae. However, since chicken manure was not used in this study, there are no concerns about the results.
4) line 307-309: “These results suggested that the microbial community composition in the hindgut of third-instar larvae of P. brevitarsis varied depending on farm conditions and might affect the microbial flora of the frass from these larvae.”
Comment: Indicate and name please what farm conditions are implied.
Response: The rearing environmental conditions due to differences in temperature, humidity, ventilation and particle size of soil are believed to affect the composition of the intestinal microorganisms of P. brevitarsis larvae. The related sentence was added in line 316.
5) line 245: one dot should be removed.
Response: The indicated error has been corrected.
6) Larval microorganisms may contain antimicrobial peptides (bacteriocins), which can affect members of human microbiological communities. It is necessary to mention this probability in the discussion and quote one of the latest publications, for example:
Simons, A.; Alhanout, K.; Duval, R.E. Bacteriocins, Antimicrobial Peptides from Bacterial Origin: Overview of Their Biology and Their Impact against Multidrug-Resistant Bacteria. Microorganisms 2020, 8, 639. https://doi.org/10.3390/microorganisms8050639
Guryanova, S.V. Immunomodulation, Bioavailability and Safety of Bacteriocins. Life 2023, 13, 1521. https://doi.org/10.3390/life13071521
Response: In “anti-bacterial and anti-inflammatory activities” in line 55, because the suggested information was included, one reference was added as “anti-bacterial and anti-inflammatory activities [17, 18].
7) In the discussion or conclusion, it is necessary to indicate what effect the microorganisms found in the larvae can have on the human body.
Response: There is no information on the direct effects of P. brevitarsis larval microorganisms on humans. However, proteins, lipids and microorganisms of P. brevitarsis larvae may have diverse pharmacological activities. Accordingly, this information is described on lines 52-55.

Reviewer 2 Report
Comments and Suggestions for Authors
The paper aims to investigate hindgut microbiota variation in third-instar Protaetia brevitarsis larvae across diverse farms through a comparative analysis based on 16S rRNA gene sequencing. The study explores the potential impact of farm-specific factors on the gut microbial communities of P. brevitarsis larvae, offering broader implications for agricultural waste recycling. The research showcases a rigorous methodology, covering thorough sampling, DNA extraction, sequencing, and bioinformatics analyses. The comprehensive exploration of hindgut bacterial communities contributes valuable knowledge to the fields of insect physiology, ecology, and sustainable waste management.
The following minor suggestions are proposed for the authors' consideration to enhance clarity, provide additional context, and guide the reader through the research:
Revise the title to "Comparative Analysis of Hindgut Microbiota Variation in Protaetia brevitarsis Larvae Across Diverse Farms."
Provide a brief mention of the methods used for gut microbiota analysis in lines 10-25 to offer readers an understanding of the techniques employed in the study. Additionally, consider elaborating on specific environmental factors influencing hindgut microbiota mentioned in the abstract.
Define or explain the terms "TO and KO farms" introduced in line 14 for better reader understanding.
In the conclusion (line 25), include a sentence or two about the potential implications of the findings for future research or applications in larval farming, traditional medicine, or biofertilizer production.
Replace the keyword “hindgut contents” with “hindgut microbiota” in line 26.
Replace the keyword “variation” with “microbiota variation” in line 26.
Elaborate on the ecological and economic significance of P. brevitarsis in lines 28-71, emphasizing its relevance to farming and potential impact on food and feed security.
Provide a brief explanation or potential reasons for significant drying yield variation among commercial farms in Korea in lines 41-42.
Rephrase lines 77-78 to "The mean larval body weight exceeded 2 grams per larva (Table 1)."
Rephrase lines 78-79 to "To obtain hindgut samples, larvae were immersed in 75% alcohol and subjected to three washes with phosphate-buffered saline (PBS) [28]."
Clarify the meaning of the "effective rate" in line 132 and provide a brief explanation for readers unfamiliar with the term.
In lines 213-227, explain what information readers can extract from the LEfSe-generated cladogram and histogram, mentioning specific insights gained from Figures 6-8.
Correct the repetition of "JH" in line 218, “… 19 from JH, 9 from KB and 24 from JH.”
Εlaborate on the "biomass decomposer role" in line 250, explaining its significance in waste reduction.
Provide a reference for the microbial sequence analysis mentioned in line 259.
Author Response
Reviewer #2: Comments and Suggestions
The paper aims to investigate hindgut microbiota variation in third-instar Protaetia brevitarsis larvae across diverse farms through a comparative analysis based on 16S rRNA gene sequencing. The study explores the potential impact of farm-specific factors on the gut microbial communities of P. brevitarsis larvae, offering broader implications for agricultural waste recycling. The research showcases a rigorous methodology, covering thorough sampling, DNA extraction, sequencing, and bioinformatics analyses. The comprehensive exploration of hindgut bacterial communities contributes valuable knowledge to the fields of insect physiology, ecology, and sustainable waste management. The following minor suggestions are proposed for the authors' consideration to enhance clarity, provide additional context, and guide the reader through the research:
Revise the title to "Comparative Analysis of Hindgut Microbiota Variation in Protaetia brevitarsis Larvae Across Diverse Farms."
Response: The title was changed.
Provide a brief mention of the methods used for gut microbiota analysis in lines 10-25 to offer readers an understanding of the techniques employed in the study. Additionally, consider elaborating on specific environmental factors influencing hindgut microbiota mentioned in the abstract.
Response: The sentence “Species richness, evenness, and diversity results using a-diversity analysis were similar between farms” was modified as “Species richness, evenness, and diversity results using a-diversity analysis (observed species, and Chao1, Shannon, and Simpson) were similar between farms” in the section of Abstract
The following sentence “Although oak sawdust and spent mushroom substrate are the main feed ingredients used on farms, the usefulness of these ingredients can be further increased by fermenting them based on the farm's know-how including special additives (e.g., rice straw, rice bran, microbial agent), temperature, humidity and mixing ratio. In addition, the rearing environment conditions due to differences in temperature, humidity, ventilation and particle size of soil are believed to affect the composition of the intestinal microorganisms of P. brevitarsis larvae” was added in line 313-319.
Define or explain the terms "TO and KO farms" introduced in line 14 for better reader understanding.
Response: Farm names were added in the footnote of Table 1 with “KB, Kyungpook farm; TO, Tohamsan gumbengi Farm; BR, Gumbengi brothers; IS, Secomnalagum gumbengi; JH, JHbio”
In the conclusion (line 25), include a sentence or two about the potential implications of the findings for future research or applications in larval farming, traditional medicine, or biofertilizer production.
Response: The sentence “it is important to first understand the composition of intestinal microorganisms of P. brevitarsis larvae for various studies such as agricultural waste recycling using gut microorganisms of P. brevitarsis larvae” was modified with “it is important to first understand the composition of intestinal microorganisms of P. brevitarsis larvae for various studies such as agricultural waste recycling, larval farming, traditional medicine or biofertilizer production using gut microorganisms of P. brevitarsis larvae” in the section of conclusion.
Replace the keyword “hindgut contents” with “hindgut microbiota” in line 26.
Response: Wording has been changed as suggested.
Replace the keyword “variation” with “microbiota variation” in line 26.
Response: Wording has been changed as suggested.
Elaborate on the ecological and economic significance of P. brevitarsis in lines 28-71, emphasizing its relevance to farming and potential impact on food and feed security.
Response: The sentence “Insect farming is one way to enhance food and feed security” was modified was “Insect farming is one way to enhance food and feed security when insects are produced in a mass production system like livestock and developed into a palatable food [2]” in line 34.
Provide a brief explanation or potential reasons for significant drying yield variation among commercial farms in Korea in lines 41-42.
Response: The addition sentence “depending on differences in the period of fasting before harvest and/or the presence or absence of fasting with water immersion” was added in line 44.
Rephrase lines 77-78 to "The mean larval body weight exceeded 2 grams per larva (Table 1)."
Response: Wording has been changed as suggested in line 80.
Rephrase lines 78-79 to "To obtain hindgut samples, larvae were immersed in 75% alcohol and subjected to three washes with phosphate-buffered saline (PBS) [28]."
Response: Wording has been changed as suggested in line 81.
Clarify the meaning of the "effective rate" in line 132 and provide a brief explanation for readers unfamiliar with the term.
Response: The word “"effective rate" was deleted in revision in line 136.
In lines 213-227, explain what information readers can extract from the LEfSe-generated cladogram and histogram, mentioning specific insights gained from Figures 6-8.
Response: The modified sentence “We performed LEfSe to generate a cladogram and histogram of LDA scores to identify microbial taxa with significant differential abundance values across different farms’ was added in line 219.
Correct the repetition of "JH" in line 218, “… 19 from JH, 9 from KB and 24 from JH.”
Response: The indicated error has been corrected.
Εlaborate on the "biomass decomposer role" in line 250, explaining its significance in waste reduction.
Response: The additional sentence “Edible insect P. brevitarsis larvae can act as important biomass decomposers, reducing waste by converting herbaceous and woody plant residues into humic acids” was added in line 254.
Provide a reference for the microbial sequence analysis mentioned in line 259.
Response: The reference [28] was added.
